# Clinical Utility of the Portable Pressure-Measuring Device for Compression Garment Pressure Measurement on Hypertrophic Scars by Burn Injury during Compression Therapy

**DOI:** 10.3390/jcm11226743

**Published:** 2022-11-15

**Authors:** So Young Joo, Yoon Soo Cho, Ji Won Yoo, Yi Hyun Kim, Rachael Sabangan, Seung Yeol Lee, Cheong Hoon Seo

**Affiliations:** 1Department of Rehabilitation Medicine, Hangang Sacred Heart Hospital, College of Medicine, Hallym University, Seoul 01000, Republic of Korea; 2Department of Internal Medicine, School of Medicine, University of Nevada Las Vegas, Las Vegas, NV 89154, USA; 3Department of Philosophy, University of Nevada Las Vegas, Las Vegas, NV 89154, USA; 4Department of Biology, College of Sciences, University of Nevada Las Vegas, Las Vegas, NV 89154, USA; 5Department of Physical Medicine and Rehabilitation, Soonchunhyang University Bucheon Hospital, Soon Chunhyang University College of Medicine, Bucheon 14584, Republic of Korea

**Keywords:** pressure measurement, burn, hypertrophic scar, pressure garment therapy, pressure-monitoring device, compression therapy

## Abstract

Compression therapy for burn scars can accelerate scar maturation and improve clinical symptoms (pruritus and pain). This study objectively verified the effect of pressure garment therapy in maintaining a therapeutic pressure range for hypertrophic scars. Sixty-five participants (aged 20~70 years) with partial- or full-thickness burns, Vancouver scar scale score of ≥4, and a hypertrophic scar of ≥4 cm × 4 cm were enrolled. Compression pressure was measured weekly using a portable pressure-monitoring device to regulate this pressure at 15~25 mmHg for 2 months. In the control group, the compression garment use duration and all other burn rehabilitation measures were identical except for compression monitoring. No significant difference was noted in the initial evaluations between the two groups (*p* > 0.05). The improvements in the amount of change in scar thickness (*p* = 0.03), erythema (*p* = 0.03), and sebum (*p* = 0.02) were significantly more in the pressure monitoring group than in the control group. No significant differences were noted in melanin levels, trans-epidermal water loss, or changes measured using the Cutometer^®^ between the two groups. The efficacy of compression garment therapy for burn-related hypertrophic scars can be improved using a pressure-monitoring device to maintain the therapeutic range.

## 1. Introduction

Hypertrophic scars caused by burns show continuation of the inflammatory reaction during the wound-healing process. This leads to complications in patients with burns not only cosmetically but also functionally. For example, such hypertrophic scars can cause joint contracture when the burn extends over a skeletal joint area. Hypertrophic scars occur after burns begin to develop in the inflammatory phase and advance to the proliferative and remodeling phases. Therefore, it is recommended that patients with burns should use a pressure garment from the start of therapy to stabilize the scar during all three above-mentioned phases. Pressure garment therapy facilitates scar maturation, impairs fibroblast activity by generating an ischemic environment, and alleviates the symptoms of pain and pruritus [1,2]. Pressure therapy has emerged as a non-invasive and cost-effective method for treating hypertrophic scars. However, the lack of standardized protocols has hindered a complete understanding of the mechanisms underlying pressure therapy [3]. Several studies have recommended a therapeutic pressure range of 15–25 mmHg to suppress scar thickness, restore vascular valve function, and prevent leg ulcers [2,4,5,6,7,8].

It is known that the compression efficiency of pressure garments decreases over time due to fabric fiber fatigue resulting from normal wear and washing [2,6,9]. The compression ability of pressure garments should be measured and adjusted periodically to maintain therapeutic functionality [10]. To address this issue, several studies have measured the pressure between the skin and the pressure garment [9,11,12,13]. The most challenging problem in designing and developing a pressure sensor is pressure measurement over the uneven surface of the scar, especially during body movements. Ideally, a pressure sensor used in the pressure-measuring device should be small, thin, and flexible to adapt to uneven body shapes [14,15,16].

Although pressure garments are widely used in compression therapy for burn-related scars, only few studies have quantified compression pressure in such patients. The researchers have developed a portable pressure-measuring device using silicon piezoresistive sensors [17]. In this study, we aimed to measure the therapeutic efficacy of treating burn-related hypertrophic scars when the compression of pressure garments was regulated and kept within the therapeutic range using a portable pressure-measuring device.

## 2. Materials and Methods

This prospective, double-blind, randomized, controlled trial enrolled eligible participants aged 20–70 years with partial- or full-thickness burns, which spontaneously healed or required a skin graft, from the Department of Rehabilitation Medicine at the Hangang Sacred Heart Hospital of Korea between October 2019 and May 2020. This study was approved by the Ethics Committee of Hangang Sacred Heart Hospital (HG 2018-026), and all patients provided written informed consent. The trial protocol has been registered at ClinicalTrials.gov (NCT04456543).

### 2.1. Clinical Participants

Sixty-five patients with hypertrophic scars from burn injuries were recruited for pressure therapy. The clinical diagnosis of hypertrophic scarring was based on standard clinical criteria. The Vancouver scar scale (VSS) was used to score the overall scar appearance. Patients with hypertrophic scars, defined as VSS score of ≥4 and located at the extremities with a size of ≥4 cm × 4 cm, were administered pressure therapy. Scars from the proliferative phase were also included immediately after epithelialization was complete. Patients with open wounds or burn scar infections, those taking steroids for the scars, those undergoing any other medical treatment, or those with underlying conditions affecting wound healing (e.g., diabetes) were excluded from the study. The participants who fulfilled our inclusion and exclusion criteria were randomly allocated to either the pressure monitoring group (n = 33) or the control group (n = 32) using a software algorithm. Two participants in the pressure-monitoring group did not undergo regular pressure monitoring, while one patient in the pressure-monitoring group and two in the control group were excluded from this study because they could not wear pressure garments for more than 8 h a day. Thus, 30 participants were included in the pressure-monitoring group and 30 in the control group (Figure 1).

### 2.2. Intervention

Patients in both the groups received standard treatment for hypertrophic scars caused by burn injuries. The standard treatment comprised occupational therapy to improve upper-limb function, physiotherapy to improve lower-limb function, stretching exercises for scar contracture, moisturizing cream, and silicone gel application. In the pressure-monitoring group, garment pressure was monitored using a portable pressure-measuring device once a week. For pressure measurement on the forearm, the patient was seated in a chair with the upper arm on a table (Figure 2). Additionally, measurements of the lower extremities were performed in the supine position. The device was attached to the patient using a Velcro belt. The sensor was placed over the scar, and a researcher held it in place while the patient wore the pressure garment. The measurement time was approximately 10 s. The interface pressures were then displayed on a computer via Bluetooth data transmission from the device and stored in a database. The visualization software presents measurements both numerically and graphically. The flexible piezoresistive sensors allowed pressure to be acquired at the pressure garment–scar interface in mmHg units. The pressure garment was adjusted weekly by tightening the loosened garment such that the pressure was maintained within the therapeutic range of 15–25 mmHg. In the control group, the standard treatment for burn scars, except for pressure monitoring, was performed in the same manner. The garment manufacturer measured the reduced body dimensions and customized the pressure garment by modifying it whenever the participants felt that the pressure garment fit over the scar area was loose. The participants were instructed to wear the garment throughout the day except when showering or managing scars.

### 2.3. Outcome Measurements

To evaluate the effect of the novel portable pressure-monitoring device, skin test results for thickness, melanin, erythema, trans-epidermal water loss (TEWL), and skin elasticity levels between the two groups were compared from baseline measurements immediately before treatment to the measurements after 2 months (Figure 3). Room temperature and relative humidity were maintained at 20–25 °C and 40–50%, respectively. With the patient in the supine position, skin thickness and other parameters were measured at the same location where the pressure sensors were attached. The primary outcome was the effect on scar thickness. Scar thickness was measured using ultrasonic wave equipment (128 BW1 Medison, Seoul, Republic of Korea). Mexameter^®^ (MX18, Courage-Khazaka Electronics GmbH, Cologne, Germany), which operates on the principle of “photo-spectrum analysis”, was used to measure the levels of melanin and the severity of erythema in the skin in arbitrary unit values ranging from 0 to 999. Higher values indicate darker and redder skin [18,19]. TEWL was measured using a Tewameter^®^ (Courage-Khazaka Electronic GmbH, Germany) to evaluate water evaporation. Elasticity was measured using a Cutometer SEM 580^®^ (Courage-Khazaka Electronic GmbH, Cologne, Germany), which applies a negative pressure (450 mbar) to the skin. The numeric values (in mm) of the skin distortion are presented as elasticity. A complete cycle comprised 2 s of negative pressure of 450 mbar followed by 2 s of recess. Three measurement cycles were conducted to obtain an average value. These parameters consisted of the following biomechanical skin properties: distensibility, elasticity, and viscoelasticity. Distensibility refers to the total displacement from the initial position at the maximum negative pressure. Gross elasticity refers to the ability of the skin to return to its initial position after displacement. Biological elasticity refers to the ratio of immediate retraction to total displacement. Viscoelasticity is defined as the ratio of delayed to immediate distension [15]. The outcome measurements and data analyses were performed by a trained and blinded outcome assessor who was not involved in the intervention.

### 2.4. Statistical Analysis

The Shapiro–Wilk test was used to determine the normal distribution of the variables. Normally distributed data were analyzed using an independent *t*-test. Non-normally distributed or non-parametric data were analyzed using the Mann–Whitney *U*-test. When comparing baseline characteristics, age, total burn surface area (TBSA), and TEWL were analyzed using independent *t*-tests. The scar thickness, melanin level, erythema severity, sebum level, Cutometer^®^ measurements, and duration between the burn injury and treatment were analyzed using the Mann–Whitney *U*-test. Fisher’s exact test was used for categorical data (sex, burn site, and burn mechanism). The amount of change (pre- to posttreatment) between the two groups was evaluated using the Mann–Whitney *U*-test for all parameters. Data analysis was performed using SPSS version 23.0 (IBM, Armonk, NY, USA). Statistical significance was set at a *p*-value < 0.05.

## 3. Results

The baseline characteristics are summarized in Table 1. No significant differences were noted between the two groups in terms of demographic characteristics or in the initial assessment of skin thickness and properties (Table 2).

The amount of change between pre-treatment and post-treatment scar thickness (*p* = 0.03) and erythema severity (*p* = 0.03) was significantly reduced in the pressure-monitoring group compared to the control group (Table 3). Sebum levels (*p* = 0.02) were significantly higher in the pressure-monitoring group than in the control group. However, no significant differences were noted in melanin levels and scar TEWL. The amount of change measured using the cutometer (skin distensibility, biological skin elasticity, gross skin elasticity, and skin viscoelasticity) showed no significant differences between the two groups.

## 4. Discussion

In this study, our data confirmed that for hypertrophic scars caused by burns, maintaining the therapeutic pressure range using a portable pressure-monitoring device suppressed scar growth and improved scar characteristics, including erythema and sebum.

Several clinical studies have reported the positive effects of compression therapy on hypertrophic scars caused by burns. Since then, efforts have been focused on accurate pressure quantification to better define the compression protocol and treatment mechanism. The mechanism of compression therapy involves regulating collagen synthesis by suppressing the vascular supply of oxygen and nutrition to the scar area [20]. The pressure-monitoring group showed improved microcirculation-related erythema and scar thickness. Other studies have shown similar results for the treatment of hypertrophic scars, wherein scar thickness and hardness were improved during compression therapy [4,21]. Previous studies have reported that scar erythema shows a high correlation with microvasculature. Hence, reduced erythema may indicate a decrease in microcirculation within the scar tissue, thereby reducing scar thickness [22]. Results from in vitro studies demonstrated that pressure-induced ischemic environments inhibit the activity of myofibroblasts and prevent scar formation [15,22,23,24]. It has been emphasized that monitoring the degree of compression during compression therapy is necessary to improve the condition of the scar area and clinical symptoms such as pain and pruritus [25]. The accuracy of pressure measurement between the pressure garment or skin/scar surface using the Oxford Pressure Monitor, PicoPress^®^, and Kikuhime (Medigroup, Melbourne, Australia), which are known to be clinically valuable as pressure-measuring devices in various diseases, has been questioned, and the need for a real-time portable pressure-monitoring device has been suggested [26,27,28]. We developed a portable pressure-monitoring device and have shown its superior reliability and validity compared to existing pressure-measuring devices [11].

Tissue engineering is a rapidly advancing multidisciplinary research field with significant promise for regenerative medicine that aims to fabricate and develop tissues to replace damaged tissues [29]. Efforts are underway for skin restoration using biomaterials and for improvement in characteristics of scars resulting from injuries to the skin, such as full-thickness burns or genetic diseases [30]. In this study, microcirculation-related factors, such as erythema, scar thickness, and sebum level, were improved. A therapeutic mechanism has been suggested wherein compression therapy accelerates scar maturation by affecting the cells around endothelial cells or glands. Experimental results on the regeneration effects during compression were confirmed by inhibiting cells such as myofibroblasts or keratinocytes that stimulate scar formation in the epidermis and dermis [22,31]. We noted that the changes in erythema and scar thickness were reduced in the pressure-monitoring group, which is similar to previous research results. However, the two groups showed no significant difference in skin properties. Scars have a lower blood supply, which results in less elasticity [32]. Scars have abnormal collagen patterns and alterations in the proteoglycan matrix [18,33]. It has been postulated that pressure therapy changes the alignment of the collagen fibers and improves the hardness of the scar by inhibiting collagen formation [5]. Pressure therapy also alleviates pruritus and pain associated with active hypertrophic scars [10,34]. Therefore, early pressure garment therapy during wound healing may control scar inflammation and proliferation and facilitate the remodeling phase to accelerate scar maturation [20]. Moreover, mechanical compression has been reported to modulate remodeling during wound healing [35]. Generally, if garment compression therapy was performed for a long duration, such as 1.5–2 years, scar thickness may be reduced. However, since hypertrophic scars in burn injuries continue to increase in thickness in the proliferative phase, a change in elasticity that can be measured with a cutometer after a short, two-month compression therapy should not be expected.

To confirm the exact therapeutic pressure range, a comparative analysis of the clinical effects at various pressure levels is required in the future. Basic research on the epidermis and scar-forming cells of the dermis as well as other adnexa organs should be performed in parallel to confirm the clinical effect as well as the clear treatment mechanism. In this study, to obtain better clinical results during compression therapy, the necessity of periodic monitoring and adjustment of the compression garment to maintain the compression pressure at an optimal therapeutic level rather than the classical method of compression treatment at monthly intervals or according to the patient’s subjective feeling of pressure was confirmed.

## 5. Conclusions

Compression treatment is an effective therapy for the clinical improvement of hypertrophic scars caused by burns. Maintaining the therapeutic range of compression through continuous pressure monitoring is essential.

## Figures and Tables

**Figure 1 jcm-11-06743-f001:**
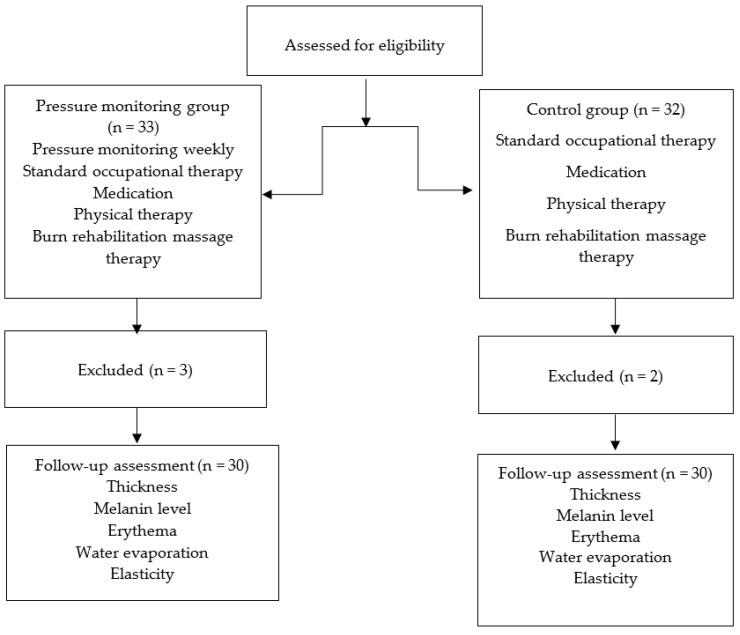
Schematic for subject enrollment, allocation, and follow up.

**Figure 2 jcm-11-06743-f002:**
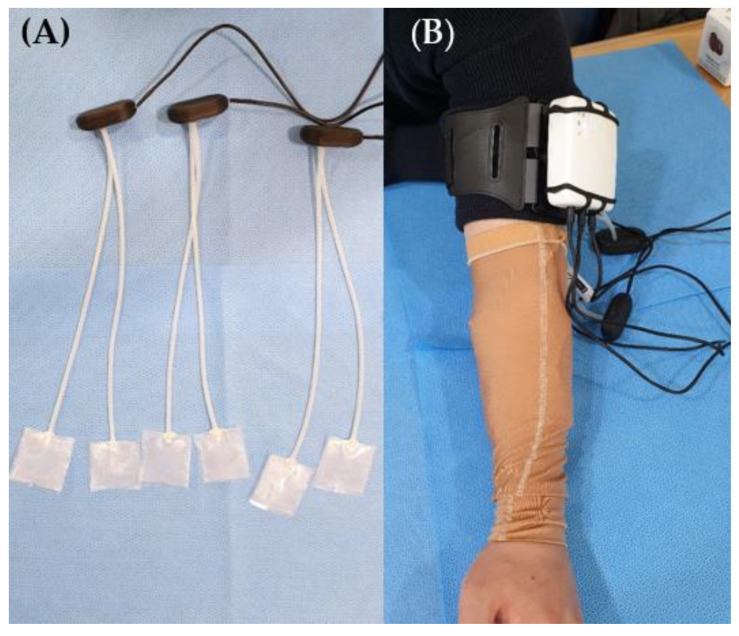
(**A**) Portable pressure-measuring device, sensor; (**B**) pressure garment placement on a patient’s forearm.

**Figure 3 jcm-11-06743-f003:**
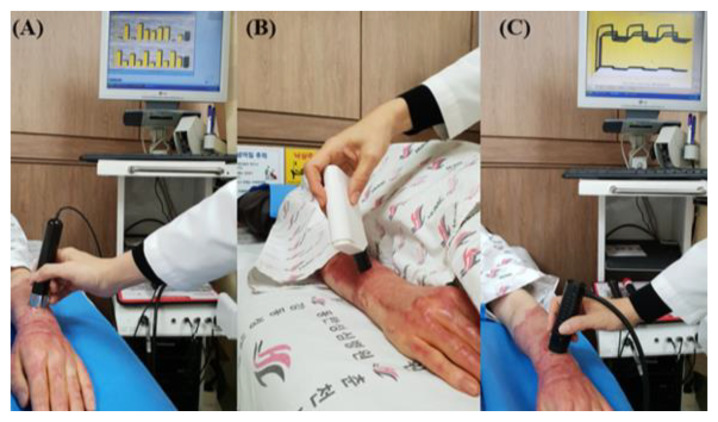
(**A**) Measurement of melanin levels and erythema severity using Mexameter^®^ (MX18, Courage-Khazaka Electronics GmbH, Germany), (**B**) measurement of sebum using Sebumeter^®^ (Courage-Khazaka Electronic GmbH, Germany), and (**C**) measurement of elasticity using Cutometer SEM 580^®^ (Courage-Khazaka Electronic GmbH, Cologne, Germany).

**Table 1 jcm-11-06743-t001:** Demographic characteristics of participants (n = 60).

	Pressure Monitoring (n = 30)	Control (n = 30)	*p*
Male:Female, n	28:2	25:5	0.62
Age (years)	44.13 ± 14.51	46.03 ± 11.78	0.58
TBSA (%)	31.93 ± 16.82	35.57 ± 14.57	0.60
Site of burn 0.70			
Arms, thigh	3 (10)	2 (7)	
Forearms, leg	15 (50)	16 (53)	
Hands, foot	12 (40)	12 (40)	
Mechanism of burn			0.65
Flame	20 (67)	18 (60)	
Electrical	7 (23)	7 (23)	
Scalding	3 (10)	5 (17)	
Duration (days) between the burn injury and treatment	69.00 ± 30.05	65.67 ± 20.00	0.90

Data are presented as mean ± standard deviation. TBSA, total burn surface area; SB, scalding burn; FB, flame burn; CoB, contact burn.

**Table 2 jcm-11-06743-t002:** Pre-homogeneity test of initial assessment.

	Pressure Monitoring (n = 30)	Control (n = 30)	*p*
Thickness (cm)	0.19 ± 0.06	0.18 ± 0.05	0.96
Melanin (AU)	185.93 ± 78.49	159.60 ± 73.18	0.84
Erythema (AU)	517.57 ± 114.30	491.43 ± 100.11	0.78
TEWL (g/h/m^2^)	17.31 ± 5.45	15.92 ± 5.86	0.89
Sebum	33.30 ± 54.93	30.57 ± 46.52	0.27
Skin distensibility	0.66 ± 0.61	0.61 ± 0.58	0.90
Biologic skin elasticity	0.43 ± 0.23	0.42 ± 0.24	0.71
Gross skin elasticity	0.60 ± 0.29	0.62 ± 0.21	0.69
Skin viscoelasticity	0.49 ± 0.44	0.45 ± 0.20	0.77

Data are presented as mean ± standard deviation; AU, arbitrary units; TEWL, trans-epidermal water loss.

**Table 3 jcm-11-06743-t003:** Change score (pre- to posttreatment) on measured outcomes.

	**Pressure Monitoring** **(n = 30)**	**Control** **(n = 30)**	** *p* **
Thickness (cm)	−0.01 ± 0.08	0.06 ± 0.10	0.03 *
Melanin (AU)	8.87 ± 86.31	15.80 ± 82.07	0.42
Erythema (AU)	−93. 73 ± 116.43	−32.73 ± 117.97	0.03 *
TEWL (g/h/m^2^)	0.89 ± 6.65	0.72 ± 7.81	0.56
Sebum	74.97 ± 80.62	36.20 ± 93.63	0.02 *
Skin distensibility	0.07 ± 0.52	−0.08 ± 0.64	0.66
Biologic skin elasticity	0.01 ± 0.15	0.00 ± 0.19	0.83
Gross skin elasticity	−0.01 ± 0.19	−0.10 ± 0.29	0.10
Skin viscoelasticity	0.02 ± 0.47	0.08 ± 0.16	0.18

Data are presented as mean ± standard deviation; * *p* < 0.05, between groups; * Mann–Whitney test; AU, arbitrary units; TEWL, trans-epidermal water loss.

## Data Availability

Not applicable.

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
