# Peer review of "Clinical Utility of the Portable Pressure-Measuring Device for Compression Garment Pressure Measurement on Hypertrophic Scars by Burn Injury during Compression Therapy"

_jcm, 2022, doi:10.3390/jcm11226743_

Round 1

Reviewer 1 Report

Deat authors, 

Your study in interesting. 

The method and material are good. 

To the results it would be interesting to show through figures (photos) the evolution and the results in the studied cases. 

Thank you!

Author Response

Reviewer 1.

Your study in interesting. 

The method and material are good. 

To the results it would be interesting to show through figures (photos) the evolution and the results in the studied cases. 

Answer> We agree with your opinions. However, we were unable to take pictures of the scar area because we did not obtain consent for the pictures from the patients before the start of the study. We added the necessity of a study on the change in scar properties according to the compression period to the limitation section in a future study.

Reviewer 2 Report

I have checked the manuscript, but I cannot understand the novelty and strength of this study clearly. Discussion contents are poor, and the references are not the newest. Due to the expectation of significant improvement, I recommend a major revision.

1.

The authors should explain the combination of regenerative therapy and measurement methods.

2.

The authors should introduce biomaterial-based therapy because the pressure measurement machine is a biomaterial system.

Diagnosis   https://doi.org/10.1016/j.biomaterials.2017.11.030

Therapy  Int. J. Mol. Sci. 202122(16), 8657

Author Response

I have checked the manuscript, but I cannot understand the novelty and strength of this study clearly. Discussion contents are poor, and the references are not the newest. Due to the expectation of significant improvement, I recommend a major revision.

  1. The authors should explain the combination of regenerative therapy and measurement methods.

The authors should introduce biomaterial-based therapy because the pressure measurement machine is a biomaterial system.

Answer> We agree with your opinions. We added the contents about regenerative therapy and biomaterial based therapy in relation to the results to the discussion section, and added details to the measurement methods in the method section. We hope that these proofs will help readers understand this article clearly.

Reviewer 3 Report

Manuscript Number: JCM #1960877

Title: Clinical Utility of the Portable Pressure Measuring Device for Compression Garment Pressure Measurement on Hypertrophic Scar by Burn Injury during Compression Therapy

Journal of Clinical Medicine

Congratulations to the authors for making progress in a challenging area of burn therapy and thank you for generally well written report.  The manuscript describes a RCT about an intervention to maintain compression therapy, however, the title and the central premise of the study seems to be the measurement device itself. The focus of the study is off point in other words and seems to be a mixture of two manuscripts – one about the reliability and validity of the device and one about the impact of maintaining adequate pressure therapy on scar. The authors must step back and reconcile the focus of this paper for the target audience.

In addition, please address the following individual comments:

Abstract

1.       Please update after addressing the comments.

2.       Please add some patient group descriptives.

Introduction

1.       Line 37 – Please adjust expression – ‘suffers’

2.       Line 47 – please correct typo – restore

Methods

3.       Please add detail to Methods. What were the specific inclusion criteria related to the time since injury (maturity age of the scar)?

4.       Were there any exclusions related to age and cause of injury?

5.       What host characteristics were collected for analysis and use as co-variates in the statistical modeling?

6.       Figure 1 is not accurate to the study as described. The authors have erroneously excluded patients who were initially recruited and allocated to a study group but did not complete the protocol as proposed. The patients that did not return to follow up should not be excluded from the study, rather they should be included in the study descriptives and analysis and missing data managed in the analysis using a sensitivity analysis or other appropriate method.

7.       It is unclear if the patients in both the control and the intervention groups had to return to the hospital for each measurement session? If so, were all control patients reviewed the same number of times as the intervention patients over the 2 month duration of the study?

8.       The authors claim that the pressure sensor provided reliable and valid measurements. How was this determined or in other words, what gold standard comparison was used to confirm the pressure measured by the device, was in fact the pressure delivered at the skin interface?

9.       Please include a picture of the device without a pressure garment obscuring the view of the size of the silicone measurement pad and preferably, showing how the malleable pad moulds to the skin contours.

10.   In Outcome measurements, Line 116, please state at the start of the sub-section, that the primary outcome of interest was the characteristics (quality domains) of the scar as measured by the various techniques.  

11.   In Outcome measurements, how was melanin assessed – presumably by hue measurements (colour) as for erythema?

12.   What were the reference values used to determine abnormality, especially for melanin and sebum?

13.   Acknowledging that this may be a prototype, can the authors give an idea of the cost to clinicians, of the device and the silicone pads?

Results

14.   Table 3 – please explain how a negative value was achieved and how that should be interpreted by the reader.

15.   There are no results provided which allow the reader to determine if the pressure garment therapy (pressure delivered) was different between the groups. This is a key aspect of the study which is omitted in terms of the confirmation of efficacy of the pressure garments; the method of retensioning each week using the objective assessment; and, the efficacy and sensitivity of the pressure measurement device.

Discussion

16.   Please adjust based on the focus of the report. If the report is to describe the benefit of objective measurement of pressure delivery each week on the maintenance of pressure therapy, then the Discussion should present that and be orientated for the clinician take home messages. If the authors seek to describe the device performance and develop a more technical paper, then so be it, however, JCM may not be the appropriate target journal in that instance.

Conclusion

17.   Adjust when focus of report is clarified.

Author Response

Congratulations to the authors for making progress in a challenging area of burn therapy and thank you for generally well written report.  The manuscript describes a RCT about an intervention to maintain compression therapy, however, the title and the central premise of the study seems to be the measurement device itself.

In addition, please address the following individual comments:

Abstract

  1. The focus of the study is off point in other words and seems to be a mixture of two manuscripts – one about the reliability and validity of the device and one about the impact of maintaining adequate pressure therapy on scar. The authors must step back and reconcile the focus of this paper for the target audience.

  Answer> We agree with your opinions. We specified the clear topic of this article in the abstract section and excluded unnecessary contents from the abstract section. We hope that these proofs will help readers understand this article clearly.

  1. Please add some patient group descriptives.

Answer> We agree with your opinions. We have added information about the inclusion criteria for the patient group.

Introduction

3.Line 37 – Please adjust expression – ‘suffers’, and  Line 47 – please correct typo – restore

Answer> We appreciate you careful advise. We changed ‘suffers’ to ‘complications’, ‘restore’ corrected spelling.

Methods

  1. Please add detail to Methods. What were the specific inclusion criteria related to the time since injury (maturity age of the scar)?

Answer> We agree with your opinions. We added have added criteria for scar maturation age. We hope that these proofs will help readers understand this article clearly.

  1. Were there any exclusions related to age and cause of injury?

Answer> We appreciate you careful advise. We specified in the methods section that we conducted the study on adult participants under 70 years of age, and conducted the study on the hypertrophic scar without criteria for the cause of the burn injury. In addition, criteria for hypertrophic scars were specified in methods section. We hope that these proofs will help readers understand this article clearly.

  1. What host characteristics were collected for analysis and use as co-variates in the statistical modeling?

Answer> We appreciate you careful advise. Since it was confirmed that there was no difference in the baseline characteristics of the both groups of participants, only the difference in the scar characteristics for the intervention was comopared.

  1. Figure 1 is not accurate to the study as described. The authors have erroneously excluded patients who were initially recruited and allocated to a study group but did not complete the protocol as proposed. The patients that did not return to follow up should not be excluded from the study, rather they should be included in the study descriptives and analysis and missing data managed in the analysis using a sensitivity analysis or other appropriate method.

It is unclear if the patients in both the control and the intervention groups had to return to the hospital for each measurement session? If so, were all control patients reviewed the same number of times as the intervention patients over the 2 month duration of the study?

Answer> We appreciate you careful advise. We added contents to the methods section about the reason for the loss of the both groups and the reason why we could not include the lost participants in the statistical analysis.

  1. The authors claim that the pressure sensor provided reliable and valid measurements. How was this determined or in other words, what gold standard comparison was used to confirm the pressure measured by the device, was in fact the pressure delivered at the skin interface?

Please include a picture of the device without a pressure garment obscuring the view of the size of the silicone measurement pad and preferably, showing how the malleable pad moulds to the skin contours.

Answer> We appreciate you careful advise. We have added information on how to measure the pressure in the intervention section, and picture of the sensor and pressure measureing device are added to Figure 2.

Prior to this study, an article that proved the validity and reliability of pressure measurement method using Picopress®, which is used as a standard for pressure measurement, and the developed pressure measurement device was added as a reference. We hope that these proofs will help readers understand this article clearly.

Lee, S.Y.; Cho, Y.S.; Joo, S.Y.; Seo, C.H. Comparison between the portable pressure measuring device and PicoPress® for garment pressure measurement on hypertrophic burn scar during compression therapy. Burns. 2021, 47, 1621–1626. DOI:10.1016/j.burns.2021.01.018.

  1. In Outcome measurements, Line 116, please state at the start of the sub-section, that the primary outcome of interest was the characteristics (quality domains) of the scar as measured by the various techniques.  

In Outcome measurements, how was melanin assessed – presumably by hue measurements (colour) as for erythema?

What were the reference values used to determine abnormality, especially for melanin and sebum?

Answer> We appreciate you careful advise. We added details about the skin test method to the outcome measurement section, and added Figure 3.

  1. Acknowledging that this may be a prototype, can the authors give an idea of the cost to clinicians, of the device and the silicone pads?

Answer> We appreciate you careful advise. We state that there is no conflict of interest.

Results

  1. Table 3 – please explain how a negative value was achieved and how that should be interpreted by the reader.

Answer> We appreciate you careful advise. This means that the thickness of the scar has decreased and the erythema has also decreased. As the scar thickness increases, the distensibility and elasticity decrease. We have added explanations to the discussion section for these contents. We hope that these proofs will help readers understand this article clearly.

  1. There are no results provided which allow the reader to determine if the pressure garment therapy (pressure delivered) was different between the groups. This is a key aspect of the study which is omitted in terms of the confirmation of efficacy of the pressure garments; the method of retensioning each week using the objective assessment; and, the efficacy and sensitivity of the pressure measurement device.

Answer> We appreciate you careful advise. We added the details to the method sections about the differences in interventions in both groups.

Discussion

  1. Please adjust based on the focus of the report. If the report is to describe the benefit of objective measurement of pressure delivery each week on the maintenance of pressure therapy, then the Discussion should present that and be orientated for the clinician take home messages.

Answer> We appreciate you careful advise. As pointed out, throughout the discussion, we revised the topic to the effect of monitoring the therapeutic range during compression therapy on clinical effectiveness. We also deleted previous studies on device validity and reliability. We hope that these revisions will help readers to clearly understand this study.

Conclusion

  1. Adjust when focus of report is clarified.

Answer> We appreciate you careful advise. We have modified the conclusion section to clarify the topic of this study.

Round 2

Reviewer 2 Report

The manuscript has been revised, but I cannot understand the novelty and strength of this study clearly.